# Variation in HIV care and treatment outcomes by facility in South Africa, 2011–2015: A cohort study

Jacob Bor[1,2,3]☯*, Anna Gage[4]☯, Dorina Onoya[2], Mhairi Maskew[2], Yorghos Tripodis[5], Matthew P. Fox[1,2,3], Adrian Puren[6], Sergio Carmona[6], Koleka Mlisana[6], William MacLeod[1,2]

1 Department of Global Health, Boston University School of Public Health, BA, United States of America, 2 Health Economics and Epidemiology Research Office, Department of Internal Medicine, School of Clinical Medicine, Faculty of Health Sciences, University of Witwatersrand, Johannesburg, South Africa, 3 Department of Epidemiology, Boston University School of Public Health, BA, United States of America, 4 Department of Global Health and Population, Harvard T.H. Chan School of Public Health, BA, United States of America, 5 Department of Biostatistics, Boston University School of Public Health, BA, United States of America, 6 National Health Laboratory Service, Johannesburg, South Africa

☯ These authors contributed equally to this work.
* jbor@bu.edu

**Data Availability Statement:** Access to primary data is subject to restrictions owing to privacy and ethics policies set by the South African Government. Requests for access to the data can

## Abstract

### Background

Despite widespread availability of HIV treatment, patient outcomes differ across facilities. We propose and evaluate an approach to measure quality of HIV care at health facilities in South Africa's national HIV program using routine laboratory data.

### Methods and findings

Data were extracted from South Africa's National Health Laboratory Service (NHLS) Corporate Data Warehouse. All CD4 counts, viral loads (VLs), and other laboratory tests used in HIV monitoring were linked, creating a validated patient identifier. We constructed longitudinal HIV care cascades for all patients in the national HIV program, excluding data from the Western Cape and very small facilities. We then estimated for each facility in each year (2011 to 2015) the following cascade measures identified a priori as reflecting quality of HIV care: median CD4 count among new patients; retention 12 months after presentation; 12-month retention among patients established in care; viral suppression; CD4 recovery; monitoring after an elevated VL. We used factor analysis to identify an underlying measure of quality of care, and we assessed the persistence of this quality measure over time. We then assessed spatiotemporal variation and facility and population predictors in a multivariable regression context.

We analyzed data on 3,265 facilities with a median (IQR) annual size of 441 (189 to 988) lab-monitored HIV patients. Retention 12 months after presentation increased from 42% to 47% during the study period, and viral suppression increased from 66% to 79%, although there was substantial variability across facilities. We identified an underlying measure of

be made via the Office of Academic Affairs and Research at the National Health Laboratory Service through the AARMS research project application portal: http://www.aarms.nhls.ac.za.

**Funding:** The authors received financial support from the following National Institutes of Health NIAID: 1R01AI115979 (MM, MPF, WM), 1R01AI152149 (JB, DO); and NICHD: 1R01HD084233 (JB), 1R01HD103466 (MM). The funders had no role in study design, data collection and analysis, decision to publish, or preparation of the manuscript.

**Competing interests:** The authors have declared that no competing interests exist.

**Abbreviations:** ART, antiretroviral therapy; CDW, Corporate Data Warehouse; NDOH, National Department of Health; NHI, National Health Insurance; NHLS, National Health Laboratory Service; SD, standard deviation; STROBE, Strengthening the Reporting of Observational Studies in Epidemiology; UHC, universal health coverage; VL, viral load.

quality of HIV care that correlated with all cascade measures except median CD4 count at presentation. Averaging across the 5 years of data, this quality score attained a reliability of 0.84. Quality was higher for clinics (versus hospitals), in rural (versus urban) areas, and for larger facilities. Quality was lower in high-poverty areas but was not independently associated with percent Black. Quality increased by 0.49 (95% CI 0.46 to 0.53) standard deviations from 2011 to 2015, and there was evidence of geospatial autocorrelation ($p < 0.001$). The study's limitations include an inability to fully adjust for underlying patient risk, reliance on laboratory data which do not capture all relevant domains of quality, potential for errors in record linkage, and the omission of Western Cape.

## Conclusions

We observed persistent differences in HIV care and treatment outcomes across South African facilities. Targeting low-performing facilities for additional support could reduce overall burden of disease.

---

## Author summary

### Why was this study done?

- Health outcomes have improved as South Africa has expanded access to HIV treatment, yet gaps in the quality of care remain.

- We develop a measure of quality of HIV care for public sector health facilities in South Africa using harmonized routine data and describe variation in quality over time and space.

### What did the researchers do and find?

- We define a measure of quality of HIV care for facilities from 2011 to 2015 using data from the National Health Laboratory Service (NHLS) Corporate Data Warehouse, which have been deduplicated to enable longitudinal patient follow-up.

- Measures of care outreach, retention, CD4 recovery, viral monitoring, and viral suppression were combined into an index using factor analysis.

- The quality measure had high reliability across years. Quality improved over the 5 years of the study and was higher in clinics relative to hospitals, in larger facilities, and in rural facilities. High-quality facilities tended to cluster near one another geographically. Quality of care was lower in high-poverty areas, but was not independently correlated with racial composition.

### What do these findings mean?

- While quality of HIV care improved in South Africa over the 5 years of the study, there was large variation in quality across facilities.

- Understanding differences in quality of care over time and across space could guide interventions to achieve better health outcomes.

## Introduction

Access to healthcare has improved in many low- and middle-income countries. However, quality of care remains highly variable. A recent Lancet report found that "poor-quality care is now a bigger barrier to reducing mortality than insufficient access" [1,2]. In South Africa, improving quality of care in the public sector is a key pillar of the country's National Health Insurance (NHI) strategy and is essential to reach universal health coverage (UHC) [3].

Widespread access to antiretroviral therapy (ART) has transformed HIV into a chronic disease [4], raised population life expectancy [5], and reduced HIV transmission [6]. Today, 96% of South Africans reside within 10 kilometers of a health facility providing ART (authors' calculations). However, there remain gaps in ART coverage among young people and men [3]. Improving quality of care will be critical to reach HIV treatment targets and further reduce HIV burden of disease [7]. HIV care involves repeated interactions with the health system, and people's experiences of high- or low-quality care may shape future care-seeking behaviors including HIV testing, ART uptake and adherence, and retention in care.

Quality is reflected in the inputs and structures of care, in the processes of service delivery, and in the resultant clinical outcomes [8]. For example, HIV treatment facilities may vary in inputs such as staffing levels, physical infrastructure and supplies, supply chains, and skilled managers. Facilities may vary with respect to processes of service delivery, such as the extent to which they provide guideline-recommended services, the availability of programs to support patient retention and adherence, and staff behavior toward patients [9]. Although inputs are often measured in administrative data systems, studies have found that inputs are poorly correlated with other measures of quality [10]. As a result, the 2018 Lancet Global Health Commission on High Quality Health Systems called for new quality measures that focus on the "processes and outcomes that matter most to people" including health outcomes and user experience [1]. Focusing on outcomes, high-quality HIV care means that the patient seeks care early in infection (at a high CD4 count), starts ART quickly, is retained in care, and achieves immune recovery and viral suppression. Outcome-based measures reflect variation both in clinical care and in patient characteristics and therefore should be interpreted differently than input-based measures. However, such measures focus attention on what really matters and offer a common benchmark to which all facilities can aspire.

In this paper, we define a single measure of quality of HIV care using an index of patient outcomes that are particularly sensitive to the quality of clinical care for facilities in South Africa. South Africa has the largest number of people living with HIV (8 million) and the largest HIV treatment program (4 million) in the world. The National Health Laboratory Service (NHLS) conducts all laboratory monitoring for the public sector HIV treatment program and curates records in a centralized database. Analyzing the NHLS database, we construct standardized, longitudinal, patient-level indicators of high-quality HIV care and treatment outcomes. These measures include median CD4 counts at presentation, retention in care among different groups, viral suppression, immune recovery, and monitoring for treatment failure. We describe variation in quality of care across facilities, evaluate the persistence of quality over time, and assess predictors of quality.

## Methods

### Ethics approval and reporting

The study was approved by the Human Research Ethics Committee of the University of the Witwatersrand and the Boston University Institutional Review Board for use of deidentified data with a waiver of consent. This study is reported following Strengthening the Reporting of Observational Studies in Epidemiology (STROBE) guidelines (S1 Checklist).

### Data source and study population

We analyzed data on all CD4 counts and HIV viral loads (VLs) from patients in South Africa's public sector HIV care and treatment program, from 2011 through 2016, with the exception of Western Cape Province. South Africa's NHLS conducts all laboratory monitoring for the national HIV program, and all test results are archived in the NHLS Corporate Data Warehouse (CDW). The data contained in NHLS's CDW are the same data used in patient care and are in fact a more complete representation of the laboratory tests conducted than those that get documented in patient charts [11].

NHLS data are archived at the level of the laboratory specimen. To enable longitudinal follow-up of individual patients within the NHLS database, we previously developed, applied, and validated a record linkage algorithm, creating a unique patient identifier. Linking together over 150 million HIV-related laboratory tests in the NHLS database since 2004, we constructed a national HIV cohort, which is described elsewhere [12]. The linkage achieved 99% positive predictive value and 94% sensitivity compared to manually coded data. Because HIV care involves routine laboratory monitoring, it is possible to observe patients' path through the treatment cascade using laboratory data, including date of clinical presentation [13], uptake of ART [14,15], retention in care [16], and viral suppression [17]. One of the benefits of the cohort is the ability to observe patients regardless of where they seek care in the public health system [16].

The NHLS National HIV Cohort includes patients receiving care at over 4,000 facilities. For the purposes of this study, we excluded all facilities with fewer than 20 laboratory-monitored HIV patients (individuals with a CD4 or VL) per year as well as a small number of facilities that had missing data on first CD4 visits or retention during a study year. Facilities in Western Cape Province were excluded due to many changes in facility codes, which made it difficult to construct accurate facility time series. A small number of CD4 counts had extreme values >5,000 cells/uL and were excluded. The resulting study population included all patients with valid CD4 count or VL measurements in 8 out of 9 South African provinces at all but the smallest facilities (**S1 Fig**).

### Measures of high-quality HIV care and treatment

We measured several dimensions of facility performance related to the longitudinal HIV care cascade [18]. While this study did not have a prespecified analysis plan, the measures were chosen a priori to reflect process and outcome measures related to the HIV care cascade observable in the laboratory data. Analyses were conducted at the level of the facility-year. However, the longitudinal nature of the underlying data enabled us to measure aspects of the clinical care cascade that require repeat patient-level measurements. For longitudinal measures such as retention, patient outcomes were assigned to the facility-year in which the interval started. With patient laboratory data from January 2011 through December 2016, we were able to define annual facility quality measures for 2011 to 2014 and for the first 6 months of 2015. For each facility, in each year, we constructed the following measures.

**Median first CD4 count.** Late presentation for HIV care is a persistent problem in South Africa, and CD4 count at presentation is a widely used measure of disease progression [13,19]. Patients who present with low CD4 counts have worse treatment outcomes [4] and spend more time at risk for onward transmission of the virus [6]. National guidelines recommend provider-initiated HIV testing for any patient setting foot in a health facility regardless of HIV symptoms [20]. Facilities also coordinate with community-based HIV testing providers to find new cases and link people who test positive to care. A high median first CD4 count would therefore be a marker for high quality because the facility has control over provider-initiated testing and can influence community-based testing and linkage to care, both of which would lead people to seek care earlier in HIV infection. We assessed the median CD4 count among patients presenting for HIV care for the first time at a given facility in a given year.

**Retention 12 months after first CD4 count (CD4 <350 versus CD4 ≥350).** Patients who present for care must be retained in order to achieve positive treatment outcomes [21]. Our analysis spans multiple treatment guidelines; however, for the majority of the study period, patients were eligible for ART if they had a CD4 count <350 cells/uL or WHO stage III/IV illness [22]. In more recent years, as eligibility expanded to <500 cells/uL in 2015 and to all patients in 2016, patients <350 have still been prioritized for faster initiation due to their worse prognosis without ART. The ability of facilities to retain patients with CD4 <350 thus speaks to the extent to which facilities minimize attrition between presentation and initiation and in the first year of treatment among those patients in greatest need. Patients presenting at higher CD4 counts may not yet have been eligible for ART and, in more recent years, may have been eligible but not as highly prioritized. Nevertheless, facilities are responsible for retaining these patients in care, including regular laboratory monitoring [21]. For each group of patients (CD4 <350 and CD4 ≥350), we assessed the proportion of patients with a follow-up laboratory test (either CD4 or VL) in the 6 to 18 months after their first CD4 count. Critically, our definitions of retention include follow-up laboratory results for patients even if they transfer or are referred to other facilities [16]. In cases of transfer, care episodes were assigned to the facility where the episode began, i.e., the sending facility, to avoid bias in retention estimates. (Because unsuccessful transfers, i.e., transfers that do not lead to retention in care, are not identified in the data and cannot be linked to a receiving facility, assigning successful transfers to receiving facilities would have led to upward bias in retention at those facilities and downward bias at sending facilities).

**Retention at 12 months among patients already in care.** Patients must be retained once established in care [23]. We assessed the proportion of patients established in either pre-ART or ART care who were retained 12 months later. We defined the group established in care as those with a CD4 or VL occurring at least 6 months after their first CD4 count. (Only the first laboratory test was used as a baseline for a given patient in a given calendar year to avoid duplication.) Patients were defined as retained in care if they had a CD4 count or VL 6 to 18 months after their baseline prevalent laboratory test. We note that this measure of retention includes people who have been on treatment for many years as well as those who entered care just 6 months before and may not yet be on treatment. Importantly, however, this measure captures aspects of retention related to continuation in care rather than HIV testing and care-seeking among people presenting for the first time.

**Viral suppression among patients monitored.** In addition to finding, initiating, and retaining patients, high-quality facilities must support patients in adhering to therapy, monitor for treatment success, and switch patients to second-line therapy if resistance develops. These aspects of quality can be measured by the proportion of patients who achieve viral suppression. We therefore assessed the proportion of patients whose first VL measurement in a given calendar year was <1,000 copies/mL. (The threshold of >1,000 copies/mL is one of the thresholds used in South Africa to diagnose treatment failure.)

**CD4 recovery.** Until recently, national guidelines specified both CD4 and VL monitoring for patients on treatment. Some facilities have been slow to implement VL monitoring and have relied instead on CD4 monitoring as an alternate measure of treatment outcomes. National guidelines specified CD4 monitoring at presentation, at treatment start, 6 months, 12 months, and annually thereafter until 2013, when CD4 counts were no longer collected after 1 year on ART. While VL monitoring was being scaled up, many facilities had already been collecting CD4 counts for years. We defined CD4 recovery as the proportion of patients whose first CD4 count in a given year was at least 100 cells/uL greater than their first CD4 count (at presentation). Most patients who initiate treatment see their CD4 count rise by at least this much in the first year on therapy [24]. Lack of CD4 recovery may signal delayed treatment initiation or inconsistent engagement in care.

**Repeat viral load within 6 months if unsuppressed.** A VL >1,000 copies/mL signals possible treatment failure. Guidelines specify enhanced adherence counseling with a follow-up VL within 3 to 6 months to determine whether in fact the patient is infected with a first-line drug-resistant viral strain, indicating a switch to second-line medications [25]. We assessed the proportion of patients with a VL >1,000 copies/mL who had a repeat VL measurement within the period 0 to 6 months.

The viral suppression and repeat VL indicators were missing in less than 10% of the study facility-years due to a phased adoption of less frequent VL monitoring and small numbers of unsuppressed patients. We imputed these values to maintain the balanced panel using a multivariate normal distribution using the other HIV quality indicators and the number of laboratory-monitored HIV patients per year.

## Facility and population characteristics

Understanding the relationship between care outcomes and facility and population characteristics can provide insights into (a) possible causal determinants of quality; (b) the extent to which quality is equitably distributed across different populations; and (c) facility performance, risk-adjusted for factors beyond the facility's control. For facility characteristics, we estimated the number of patients in HIV care as the total number of unique patients with either a CD4 or VL result at a given facility in a given year. We also extracted information on facility type from Department of Health records, distinguishing between primary health clinics, district hospitals or community health centers, and provincial or national hospitals.

We estimated population characteristics at the local municipality level from 2011 Census microdata (10% sample) [26]. We identified municipalities using facility geographic coordinates obtained from National Department of Health (NDOH), an NDOH-NHLS facility crosswalk we developed, and the municipality shape files from the 2011 Census. We mapped 2,962 facilities to 207 municipalities. Facilities that could not be mapped were excluded from our predictive models. To examine potential inequities in quality between relevant subgroups, we included the following municipality characteristics: rural municipalities, proportion of households in poverty (below ZAR 501 per person per month or $2 per day) [27], proportion of households that identify as majority Black (i.e., "black African"), and proportion of individuals over age 60. We also present an expanded model with the additional predictors: proportion of households that had moved to the current municipality in the past 5 years, proportion of households with access to piped water, proportion of households with electricity, and proportion of individuals who were literate.

## Analytic approach

**Factor analysis.** A central contention of our paper is that patient outcomes across different dimensions of the HIV care cascade may reflect an underlying measure of facility-specific

quality-of-care. We therefore assessed the correlation structure of the care cascade measures described above and conducted factor analysis to assess for the existence of a common latent factor reflected in these measures. Because the goal was to assess correlations between different cascade measures for a given facility-year, the facility-year data were treated as independent observations in this step of the analysis. We extracted the first factor, i.e., the latent construct that explained the greatest amount of variance in the different cascade measures, and report the factor loadings. We standardized the first factor by dividing by the global standard deviation. We interpret this first factor as a measure of quality$_{it}$ for facility $i$ in year $t$.

**Reliability of the quality measure.** Reliability is the extent to which a measure captures an underlying signal, not just random noise. The availability of multiple years of data for each facility enables assessment of the persistence of quality$_{it}$ over time. First, we assessed the correlation between quality$_{it}$ and quality$_{it+1}$ across all years of data, stratifying by facility size, given that smaller facilities will exhibit greater noise. Second, taking quality as a fixed facility-level characteristic across this 5-year period, we conducted a one-way analysis of variance to assess the proportion of the variance in quality that exists at the facility level and the proportion of variance that is within facilities across years. We computed the average facility quality score across the whole period, quality, and assessed the reliability of this measure using the Spearman–Brown formula. Reliability is interpretable as the proportion of total variance in the observed measure that would be explained by the (unobserved) true measure. Reliability increases with the strength of the underlying signal in the data and decreases with the level of noise in the estimates.

**Predictors of quality and risk-adjusted quality scores.** We assessed predictors of quality$_{it}$ in a regression context. We included predictors across 4 domains: location (province, urban/rural), time (year), facility characteristics (log-patient load, facility type), and municipality population characteristics. We conducted a mixed effects multilevel regression analysis to assess what factors predict variation in quality$_{it}$. Our mixed effects models, estimated at the facility-year level, additionally included facility random effects, municipality random effects, and autoregressive error terms over time to account for the longitudinal nature of the data. These models implicitly allow for the residual variance to differ across facilities of different sizes, shrinking facility estimates toward their predicted values to a greater or lesser extent based on observed within-facility variance. As a robustness check, we modeled the autoregressive error term with the variance stratified by deciles in facility size.

We estimated separate mixed effects models including each of the predictors one by one and in a multivariable mixed effects model. Using this multiply adjusted model, we predicted the facility random effects and used these as risk-adjusted quality scores, quality$_{i,adjusted}$, i.e., facility quality adjusted for systematic variation in observed facility and population covariates.

Finally, we mapped the geographic distribution of quality, quality$_{i,adjusted}$, and changes in quality, and assessed for geospatial autocorrelation using Moran's I. To plot geographic variation, we smoothed quality using inverse distance weighting interpolation (Quantum 2.14 GIS), with a distance coefficient of 7. Inverse distance weighting creates a smoothed surface with nearby facilities contributing more information than distant facilities [28].

## Results

Our analysis included data on 3,265 facilities with observations for each year, 2011 to 2015. The median (IQR) annual size of facilities was 441 (189 to 988) lab-monitored HIV patients. **Table 1** presents the unweighted mean and standard deviation (SD) for each of the facility care cascade measures for each year. Median CD4 count at presentation increased from a mean of 303 (SD = 57) cells/mm$^3$ in 2011 to a mean of 312 (SD = 47) cells/mm$^3$ in 2015, as the

Table 1.  HIV care quality indicators in study sample of 3,265 facilities[a].

| Year | Median first CD4 count | | Retention after first CD4 0–350 | | Retention after first CD4 350+ | | Retention starting 6 mo after CD4 | | Viral suppression | | CD4 recovery | | Monitoring after unsuppressed | |
|---|---|---|---|---|---|---|---|---|---|---|---|---|---|---|
| | Mean | SD | Mean | SD | Mean | SD | Mean | SD | Mean | SD | Mean | SD | Mean | SD |
| 2011 | 303 | 57 | 0.42 | 0.13 | 0.36 | 0.14 | 0.78 | 0.09 | 0.66 | 0.26 | 0.39 | 0.17 | 0.23 | 0.22 |
| 2012 | 296 | 53 | 0.42 | 0.14 | 0.35 | 0.13 | 0.77 | 0.08 | 0.73 | 0.19 | 0.49 | 0.15 | 0.24 | 0.19 |
| 2013 | 299 | 52 | 0.44 | 0.15 | 0.36 | 0.13 | 0.78 | 0.09 | 0.74 | 0.15 | 0.59 | 0.13 | 0.28 | 0.19 |
| 2014 | 301 | 50 | 0.46 | 0.15 | 0.38 | 0.14 | 0.77 | 0.09 | 0.76 | 0.14 | 0.62 | 0.13 | 0.31 | 0.18 |
| 2015 | 312 | 47 | 0.47 | 0.19 | 0.41 | 0.17 | 0.73 | 0.13 | 0.79 | 0.12 | 0.60 | 0.14 | 0.38 | 0.18 |

[a]For all variables but median first CD4 count, the mean and standard deviation are of the facility proportion. Estimates are unweighted.

worst performing facilities improved. From 2011 to 2015, the average proportion of patients retained 12 months after presentation increased from 42% (SD = 13%) to 47% (SD = 19%) for patients with a first CD4 count <350, and from 36% (SD = 14%) to 41% (SD = 17%) for patients with a first CD4 count ≥350. Prevalent retention among those already in care fell modestly, from 78% (SD = 9%) to 73% (SD = 13%), though there was less variation across facilities than in other measures. Viral suppression increased from 66% (SD = 26%) to 79% (SD = 12%), and CD4 recovery from 39% (SD = 17%) to 60% (SD = 14%). The average proportion of patients monitored after an unsuppressed VL increased from 23% (SD = 22%) to 38% (SD = 18%) over the period of study. S1 Table shows means and SDs for the unbalanced panel including all facilities regardless of whether they had data for all years. In summary, the data reveal substantial variation in the underlying cascade measures across facilities, as well as large increases in viral suppression, CD4 recovery, and monitoring after potential viral failure over time.

The individual facility-year-level care cascade measures exhibited low-to-moderate correlations (Table 2, S2 Table). Our factor analysis yielded a primary factor with an eigenvalue of 1.517, and second and third factors with eigenvalues of 0.644 and 0.045, respectively. Higher factors had negative eigenvalues (S3 Table). Eigenvalues greater than 1 contain more information than the individual component variables. We therefore extracted only the first factor as our measure of quality of care. We note substantial residual variation in the component measures not explained by the factors, as evidenced by high "uniqueness" values in S3 Table. However, it is important to stress that this unexplained variation includes substantial random noise due to small facility populations and does not therefore imply that the first factor was unimportant. By leveraging the covariance of different facility indicators, we are able to extract a

Table 2.  Quality indicator correlations and first factor loadings for 3,265 facilities.

| | Median first CD4 count | Retention after first CD4 0–350 | Retention after first CD4 350+ | Retention starting 6 mo after CD4 | Viral suppression | CD4 recovery | Factor 1 loadings |
|---|---|---|---|---|---|---|---|
| Median first CD4 count | 1 | | | | | | −0.046 |
| Retention after first CD4 0–350 | 0.042 | 1 | | | | | 0.675 |
| Retention after first CD4 350+ | 0.093 | 0.511 | 1 | | | | 0.580 |
| Retention starting 6 mo after CD4 | −0.057 | 0.508 | 0.420 | 1 | | | 0.637 |
| Viral suppression | −0.162 | 0.157 | 0.080 | 0.166 | 1 | | 0.343 |
| CD4 recovery | −0.212 | 0.109 | 0.055 | 0.099 | 0.379 | 1 | 0.269 |
| Monitoring after unsuppressed | 0.002 | 0.197 | 0.150 | 0.248 | 0.241 | 0.151 | 0.357 |

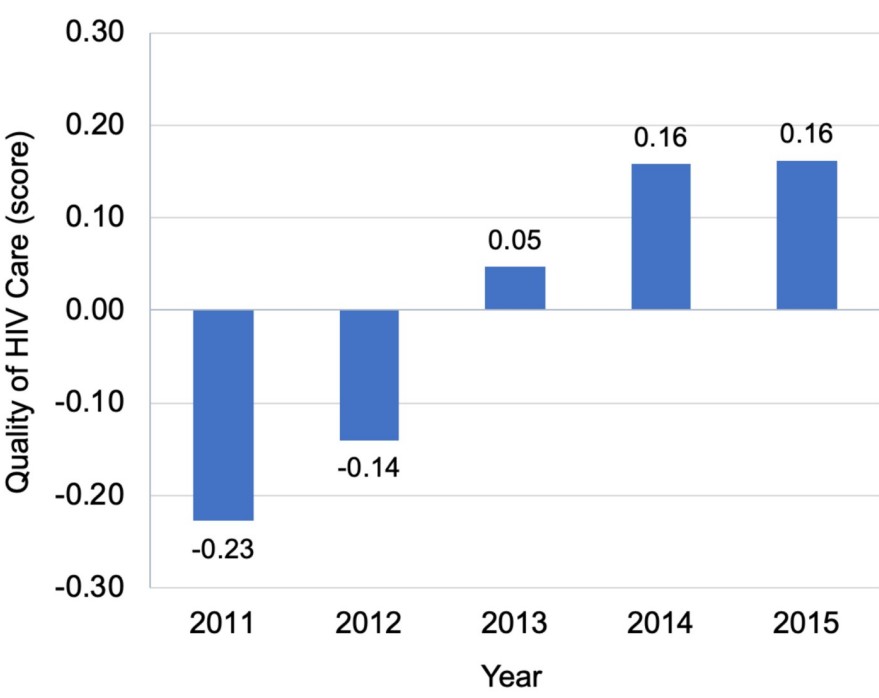

**Fig 1. Quality of HIV care across all facilities, 2011 to 2015.** Figure displays average quality scores for each year, 2011 through 2015, pooling across all facilities.

facility-level signal from the underlying data. Factor loadings on the first factor revealed moderate-to-high positive correlations with the 3 retention measures and low-to-moderate positive correlations with viral suppression, CD4 recovery, and viral monitoring if unsuppressed (**Table 2**). The first factor was not correlated with median first CD4 count, suggesting our quality measure captures aspects of care for patients already in care, but does not capture the extent to which facilities are able to screen patients for HIV earlier in infection. In contrast to the first factor, the second factor did not correlate with the underlying variables in a way predicted by theory. Therefore, we interpret only the first factor as our measure of quality of care.

Quality increased significantly over time (**Table 1**, **Fig 1**), with the average quality score rising by half a standard deviation (0.49, 95% CI 0.46 to 0.53) from 2011 to 2015. However, facility differences in quality persisted year to year. **Fig 2** displays the correlation between $quality_{i,t}$ and $quality_{i,t+1}$ in the following year for each facility-year, stratified by facility size. The year-on-year correlation is 0.70, ranging from 0.59 in the smallest size tertile to 0.83 in the largest size tertile. In an analysis of variance, 52% of the total variation was at the facility level and 48% of the variation was within-facility. When we averaged across years to obtain a single quality score for each facility, $quality_{i,crude}$, the estimated reliability of this measure was 0.84.

**Fig 3** shows how the top and bottom quintile of facilities performed on the measured quality indicators that formed the basis for the quality score. There were large differences on all measures except for first CD4 counts (as expected). However, we note that there was substantial room for improvement even for the best performing facilities.

In summary, our analysis revealed a single factor with positive loadings from all component measures chosen a priori to reflect quality of HIV care, with the exception of median first CD4 count. This factor was correlated over time, indicating the presence of a persistent "facility

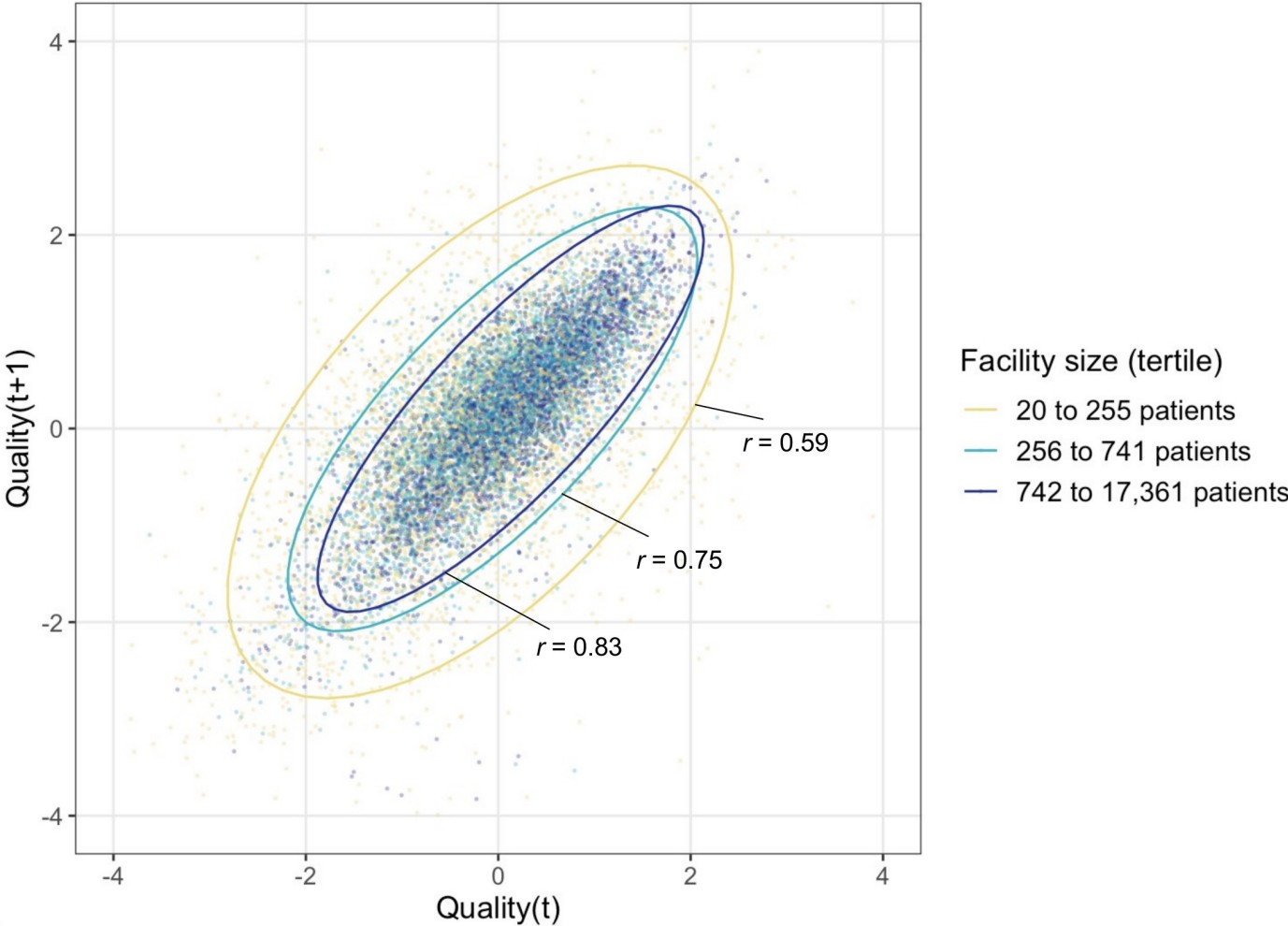

**Fig 2. Correlation of facility-specific quality scores over time, stratified by facility size.** Each data point plots a facility's quality score in year t against the quality score at the same facility in the year t+1. The plot is stratified by facility size to illustrate the greater variance in smaller facilities. Ellipses are estimated using R's stat_ellipse command [33]. Reliability within facility was 0.842, computed using Stata's loneway command, which fits one-way analysis of variance (ANOVA) models.

effect," and the mean facility quality score attained high reliability. We now turn to our assessment of predictors associated with quality of care across facilities.

How did quality vary with facility and municipality population characteristics? **S4 Table** shows summary statistics for population characteristics. Turning to our regression models (**Table 3**), we found that quality was higher in larger facilities, rising by 0.12 (95% CI 0.10 to 0.14) per log-point in facility size. This corresponds to a 0.28 SD increase in quality for a 10-fold increase in facility size, e.g., from 100 to 1,000 patients. Though larger facilities performed better on average, hospitals performed about 1 SD worse than clinics on our quality measure: −0.93 for district hospitals and community health centers and −1.13 for provincial and national hospitals. This may be explained by lower patient retention at hospitals, e.g., due to higher mortality rates among sicker patients or due to unsuccessful down-referrals to other facilities among stable patients.

Rural facilities performed better than urban facilities on our quality measure by 0.13 SD. The proportion of the population over age 60 was a strong predictor, with a 1-SD change

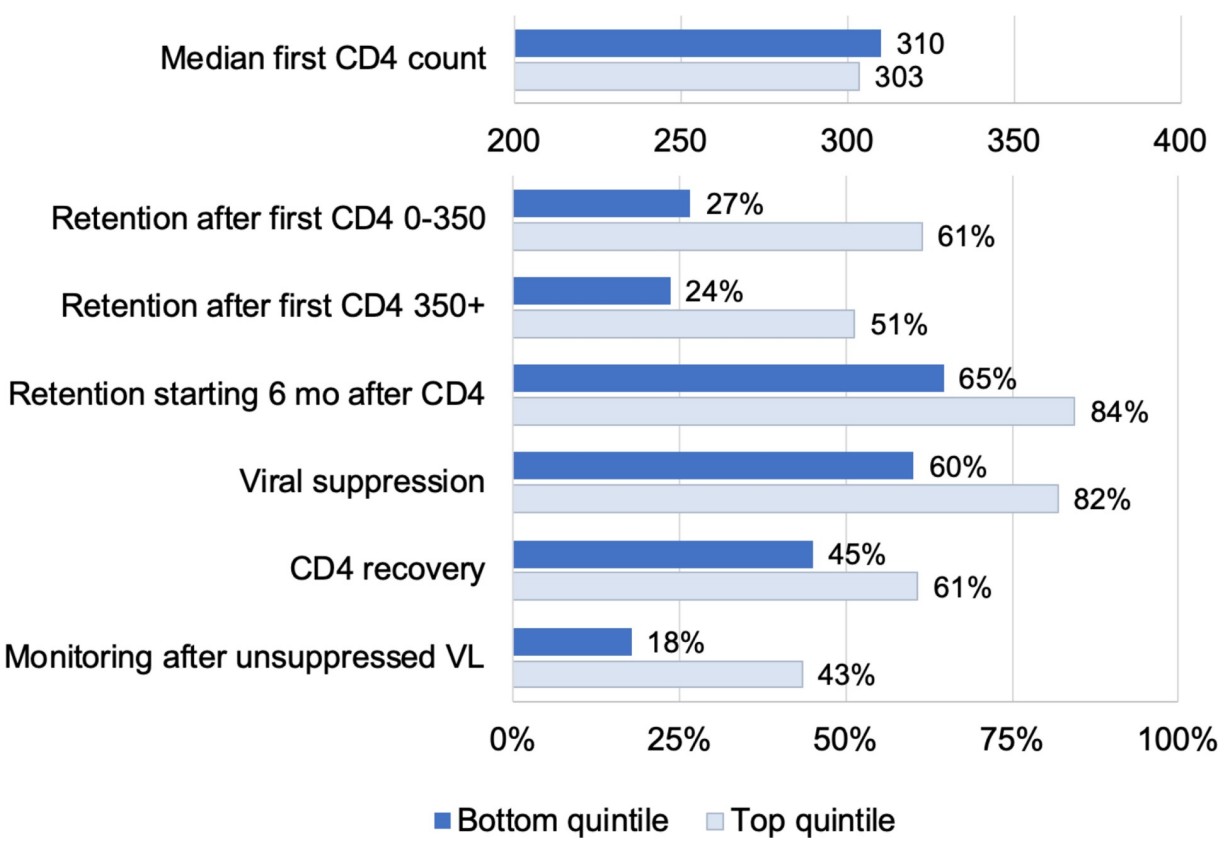

**Fig 3. Quality of care at facilities in the top (light blue) versus bottom (dark blue) quintiles.** The figure displays HIV care quality indicators for facilities with quality scores in the top 20% and bottom 20% of the distribution. All indicators are estimated from the NHLS Cohort and are described in the text.

**Table 3. Predictors of HIV care quality, 2,962 facilities.**

| | Bivariate models | | Multivariable model[a] | |
|---|---|---|---|---|
| | Beta | 95% CI | Beta | 95% CI |
| Facility characteristics | | | | |
| log-N patients | 0.14 | (0.11,0.16) | 0.12 | (0.10,0.14) |
| Facility type (clinic ref.) | | | | |
| District hospital or CHC | −0.83 | (−0.92,−0.75) | −0.93 | (−1.01,−0.84) |
| Provincial or national hospital | −1.08 | (−1.20,−0.96) | −1.13 | (−1.25,−1.01) |
| Municipality characteristics | | | | |
| Rural | 0.15 | (0.07,0.24) | 0.13 | (0.06,0.21) |
| % households in poverty[b] | 0.02 | (−0.04,0.09) | −0.10 | (−0.17,−0.02) |
| % majority Black households[b] | 0.02 | (−0.03,0.07) | −0.03 | (−0.10,0.04) |
| % population over 60[b] | 0.04 | (−0.03,0.10) | 0.17 | (0.11,0.22) |
| Constant | | | −1.05 | (−1.23,−0.87) |

CHC, community health center.

[a]Multivariable model also includes province and year fixed effects.

[b]Standardized variables.

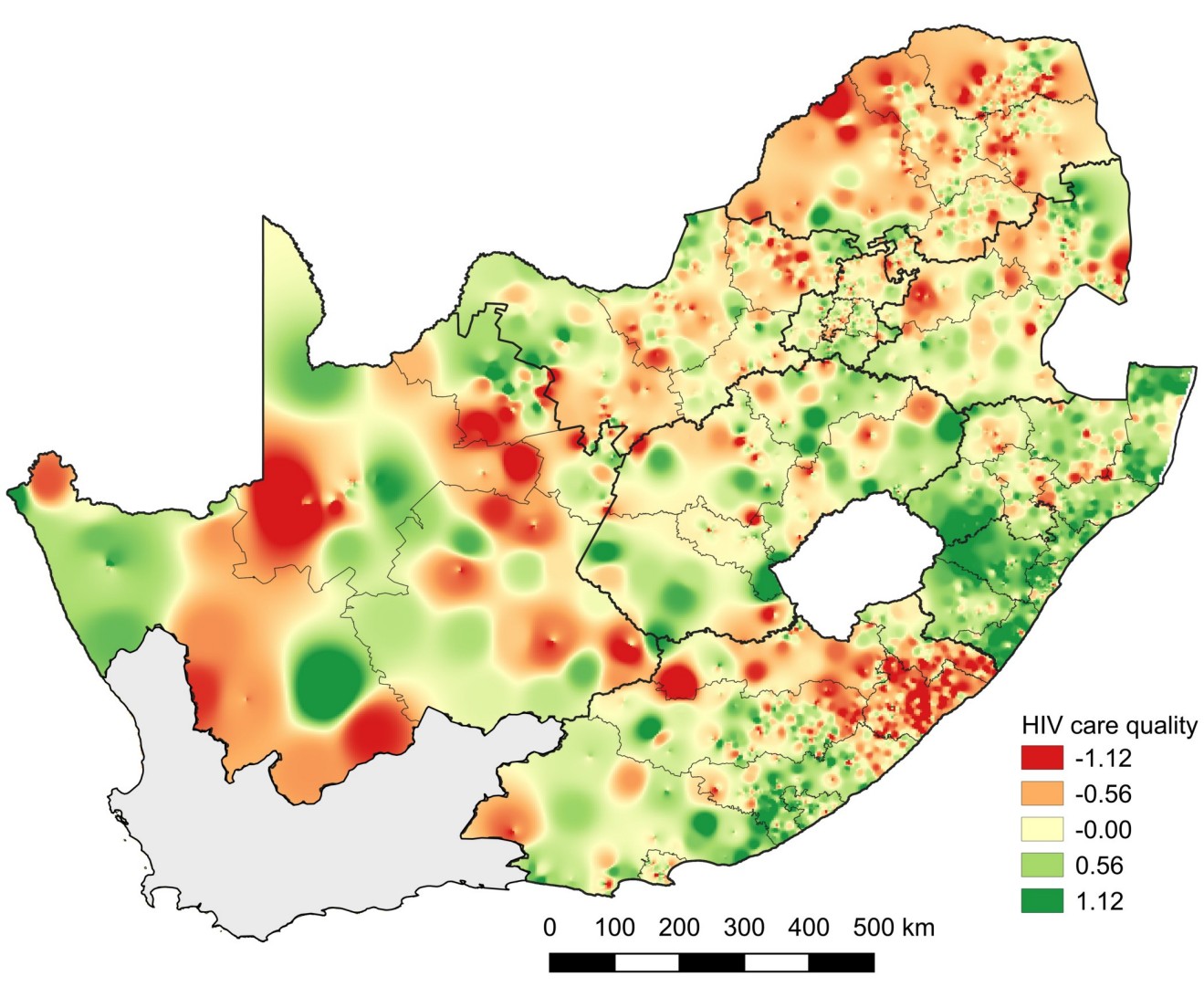

**Fig 4. Geographic variation in quality of care, 2011–2015.** Figure displays geographic variability in facility-based quality of care scores estimated for the period 2011–2015. Base map was obtained from the South African Municipal Demarcations Board (http://www.demarcation.org.za/).

associated with a 0.17 SD difference in quality. Poorer municipalities had lower quality of care, with a 1-SD increase in the poverty rate associated with a −0.10 SD reduction in quality of care. Conditional on area poverty and other predictors, percent Black was not significantly associated with lower quality. Results were similar in models including additional population predictors (**S5 Table**) and when restricting the data to a 4-year balanced panel (**S6 Table**).

Across provinces, KwaZulu-Natal had the highest average quality, Limpopo the lowest. **Fig 4** maps geographic heterogeneity in quality of care, pointing toward substantial variation and significant spatial autocorrelation ($p < 0.001$). Spatial variation in adjusted quality scores and change in quality of care at different facilities from 2011 to 2015 are shown in **S2 Fig**.

## Discussion

We developed a measure of quality of care within South Africa's public sector HIV program, based on health system-sensitive outcomes assessed through routine laboratory monitoring (2011 to 2015). This quality measure was correlated with retention in care, laboratory monitoring, CD4 recovery, and viral suppression, and it showed high stability over time (reliability = 0.84). Quality of care has improved in recent years, although there was substantial heterogeneity across facilities and there is evidence that quality is clustered geographically across space. Quality of care was negatively correlated with municipality poverty rates, but not independently associated with the proportion of the municipality that was Black, two axes of particular concern when thinking about health inequities in South Africa. Interestingly, quality was highest at primary care clinics in rural areas, perhaps because the patient population in these areas is more stable.

Our outcome-based quality measure complements ongoing efforts by South Africa's NDOH to assess variation in facility inputs and procedures [29]. Understanding differences in quality of care across time and space could guide interventions to improve quality and achieve better and more equitable health outcomes. High-quality HIV care involves screening people for HIV early in infection, starting them on treatment, retaining them in care, monitoring patients for treatment failure, and supporting patients with long-run adherence, leading to improved immune function and viral suppression. Although losses across this care cascade reflect both patient and facility-level factors, facilities can provide the high-quality care by meeting patients where they are and addressing barriers to care that they face. Improving quality of HIV care—i.e., performance along the care cascade—has been identified as the key step toward epidemic elimination [30]. Facilities could improve their quality by improving these underlying components. For example, in the highest quintile of facilities, 82% of patients achieved viral suppression, whereas 60% were suppressed in the lowest quintile. If all facilities achieved viral suppression at levels equal to facilities in the highest quintile of quality, overall suppression would be increased by 22 percentage points in the lowest quintile and by 3.5 percentage points nationally—from 81.5% to 85.0% suppressed. There has been growing interest in targeting the right programs to the right people in the right places [31]. Identifying which facilities are doing well and which are not is a key step in closing gaps in the HIV care cascade.

This analysis makes several contributions to the literature. First, we propose a summary measure of quality that incorporates performance across different dimensions of the care cascade. While the underlying data on facility-level care cascades are noisy, we identify a fair amount of stability in the first factor from our factor analysis. This measure has advantages over the common 90–90–90 cascade for measuring facility-level quality. It can be measured without facility catchment area prevalence estimates and incorporates longitudinal rather than cross-sectional estimates that better reflect the structure of chronic disease management. Second, our analysis enabled us to assess correlations between different component cascade measures. The fact that we see consistent positive correlations across most measures suggests we are picking up a facility-specific quality signal. The absence of any correlation between first CD4 and later measures may reflect lack of emphasis on case-finding, even at high-quality facilities. Facilities may view their responsibilities as starting when a person presents for care, with HIV testing relegated to partner organizations. Nevertheless, in the age of universal test-and-treat, the health system has a role to play in bringing people into care early in infection. Third, we demonstrate that quality varies systematically with a number of predictors that can guide future investments, and we identify spatial clusters of high and low quality for further investigation. Fourth, in contrast to other efforts such as the Ideal Clinic initiative [29] which focuses on inputs and processes, our quality measure focuses on proximal patient outcomes

related to the HIV care cascade that are a function of facility processes and therefore more sensitive to quality of care than distal outcome measures such as mortality or confidence in the health system. Focusing on patient outcomes is important because it reflects the health system's central purpose to improve health and it allows for flexibility in the mix of interventions that facilities use in order to achieve high-quality care [1].

Our analysis had some limitations. First, some variation in our facility-specific quality measure may reflect random fluctuations rather than underlying quality of care. However, by extracting a common factor from several underlying measures, and averaging over several years of data, we attained a measure with high reliability, indicating evidence of a persistent facility effect. Second, our measures of performance across the HIV care cascade are affected both by facility inputs and procedures as well as by characteristics of the patient population and the obstacles they may face to staying in care. We present quality estimates adjusted for population and facility characteristics; however, even these risk-adjusted quality estimates may not full adjust for important differences in patient populations across facilities and for differences in facility types. For example, if highly motivated patients choose higher-quality facilities, then our measures may reflect some element of patient selection. Third, as we have excluded very small facilities and facilities in the Western Cape, our findings are not generalizable to these facilities. We found that large facilities performed better than small facilities; however, this may not apply to very small facilities and other methods may be needed to account for the very large heterogeneity seen in those facilities. We also only use laboratory data through December 2016 before universal test-and-treat was implemented and do not yet have enough data post-implementation for comparison. Fourth, there may be changes in the municipality characteristics over the study period that are not captured in the 2011 Census. Use of the municipality data also assumes that the population use a facility in their municipality and do not travel long distances for care.

Fifth, our quality measure is based on observations of the longitudinal care cascade from the perspective of routine laboratory data, which we have linked, creating a validated unique patient identifier and national HIV cohort [12]. Although no other database has the ability to track patients through the entire health system, to observe patterns in first CD4 counts, and to track all CD4 and VL measurements, the database has limitations. Although laboratory monitoring of all patients is specified in national guidelines, gaps in monitoring may shape the data we have. Compliance with VL monitoring guidelines has increased over time, and some of the secular improvement observed may be related to these changes. We note that routine viral monitoring is a key component of high-quality care. However, we are unable to distinguish gaps in lab monitoring from other dimensions of quality in these data. Additionally, though our record linkage attained high sensitivity and positive predictive value, linkage errors may influence our results. Sixth, because our quality measure is based on longitudinal progression through care, a follow-up period is required, and quality is therefore observed with a lag.

Finally, laboratory results are but one vantage point on quality of care. Because we do not observe more frequent clinic visits or pharmacy pickups, we may miss patterns where patients cycle in and out of care between laboratory results. Future work could combine these laboratory measures with input, process, and outcome-based measures from other databases, to provide a more holistic picture of quality of care.

In identifying a persistent signal of quality of care at different facilities, this research raises a number of questions for further study. Future research should assess what facility inputs and processes are associated with higher quality; whether patients "vote with their feet," moving from facilities providing lower to higher quality of care; and what impact quality differentials have on population health outcomes. Further linkages of laboratory data with clinical and pharmacy data could yield further quality indicators—such as visit and medication pickup

adherence—and could improve the accuracy and precision of the overall quality-of-care score. Additionally, information on performance could be relayed to facilities, to help facility staff to understand where they stand relative to other facilities or relative to their past performance, and to raise expectations for performance. Sharing information with facilities on overall quality, on the component measures, and on changes over time may be an effective starting point for quality improvement. Finally, South Africa—and other countries—have made significant changes to their treatment programs in recent years with the advent of universal test-and-treat, which may affect measurement of quality of care. After September 2016, national guidelines specify that CD4 counts should still be taken at clinical presentation to assess for potential treatment complications and to target additional care to patients with advanced disease. However, because CD4 counts are no longer used to determine treatment eligibility, facilities may be less rigorous in collecting these laboratory results. Additionally, CD4 counts are no longer collected after initiation of treatment, preventing assessment of CD4 recovery. As models of care shift, the quality score will have to be updated to accommodate these changes.

With near-universal access to HIV treatment, quality of care is likely to be among the most significant factors shaping the future course of local HIV epidemics and burden of disease. In South Africa, an estimated 92% of medically preventable HIV deaths are due to poor quality of care and just 8% from lack of access [2]. Facilities that successfully diagnose new HIV infections, link people with HIV to care, retain them on therapy and facilitate transfers when necessary, and monitor them for viral suppression will improve their quality and likely see HIV incidence fall in the communities they serve. Facilities with poorer performance can expect persistent transmission, higher morbidity and mortality, and less financial protection for households [32]. Measuring differences in quality of care across facilities is a critical step toward quality improvement.

## Supporting information

**S1 Checklist. Strengthening the Reporting of Observational Studies in Epidemiology (STROBE) checklist.**
(DOC)

**S1 Fig. Exclusion flowchart.** Figure shows how the sample of facilities used in the analysis was identified after applying exclusion criteria.
(PDF)

**S2 Fig. Geographic variation in adjusted quality score (left) and changes in quality over time (right), 2011–2015.** Figure displays heat maps showing the covariate-adjusted mean quality score and changes in the quality score over time.
(PDF)

**S1 Table. HIV quality indicators for all observed facilities.** Provided as a complement to Table 1, this Supporting information table shows means and standard deviations for the unbalanced panel including all facilities regardless of whether they had data for all years.
(PDF)

**S2 Table. HIV quality indicators using 4-year panel of 3,440 facilities, 2012–2015.** Provided as a complement to Table 2, this Supporting information table displays the correlation matrix between the underlying indicators and the summary quality measure for those facilities that are observed for 4 years (rather than 5 years), enabling inclusion of additional facilities.
(PDF)

**S3 Table. Factor loadings of HIV care quality in 3,253 facilities.** Table displays factor loadings of the underlying indicators with the top 3 factors, the eigenvalues of those 3 factors, and the uniqueness of the indicators—i.e., the residual variance not explained by the factors. (PDF)

**S4 Table. Characteristics of 207 municipalities.** Table displays descriptive statistics for the municipalities where the facilities in the analysis were located. (PDF)

**S5 Table. Full model predictors of HIV care quality, 2,962 facilities.** Table displays an expanded predictive model as a supplement to Table 3. Some facilities were not included in the predictive model due to missing data on predictors. (PDF)

**S6 Table. Predictors of HIV quality using 4-year panel of 3,116 facilities.** Table displays an alternate predictive model estimated for facilities that are observed for 4 years, enabling inclusion of additional facilities. Some facilities were not included in the predictive model due to missing data on predictors. (PDF)

## Acknowledgments

The authors thank Dr. Margaret Kruk and members of the Lancet Global Health Commission on High Quality Health Systems for feedback on this work. The authors additionally thank the staff of the NHLS Corporate Data Warehouse for data management and extraction. All errors and omissions are our own.

## Author Contributions

**Conceptualization:** Jacob Bor, Anna Gage, Dorina Onoya.

**Data curation:** Jacob Bor, Adrian Puren, Sergio Carmona, Koleka Mlisana, William MacLeod.

**Funding acquisition:** Mhairi Maskew, Matthew P. Fox.

**Investigation:** Anna Gage.

**Methodology:** Jacob Bor, Anna Gage, Yorghos Tripodis.

**Supervision:** Jacob Bor, Sergio Carmona, William MacLeod.

**Visualization:** Anna Gage.

**Writing – original draft:** Jacob Bor, Anna Gage.

**Writing – review & editing:** Jacob Bor, Anna Gage, Dorina Onoya, Mhairi Maskew, Matthew P. Fox, Adrian Puren, Sergio Carmona, Koleka Mlisana, William MacLeod.

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
