## [Editor Report · Decision Letter 0]

15 Jul 2020

Dear Dr Bor, 

Thank you for submitting your manuscript entitled "Variation in HIV care and treatment outcomes across facilities in South Africa: evidence from a national laboratory cohort" for consideration by PLOS Medicine.

Your manuscript has now been evaluated by the PLOS Medicine editorial staff and I am writing to let you know that we would like to send your submission out for external peer review.

Kind regards,

Richard Turner, PhD,

Senior editor, PLOS Medicine

rturner@plos.org

---

## [Decision Letter · Decision Letter 1]

11 Aug 2020

Dear Dr. Bor,

Thank you very much for submitting your manuscript "Variation in HIV care and treatment outcomes across facilities in South Africa: evidence from a national laboratory cohort" (PMEDICINE-D-20-03265R1) for consideration at PLOS Medicine. 

Your paper was evaluated by an academic editor with relevant expertise and sent to independent reviewers, including a statistical reviewer. The reviews are appended at the bottom of this email and any accompanying reviewer attachments can be seen via the link below:

[LINK]

In light of these reviews, we will not be able to accept the manuscript for publication in the journal in its current form, but we would like to invite you to submit a revised version that addresses the reviewers' and editors' comments fully. You will recognize that we cannot make a decision about publication until we have seen the revised manuscript and your response, and we expect to seek re-review by one or more of the reviewers. 

We hope to receive your revised manuscript by Sep 01 2020 11:59PM. Please email us (plosmedicine@plos.org) if you have any questions or concerns.

Please let me know if you have any questions. Otherwise, we look forward to receiving your revised manuscript soon. 

Sincerely,

Richard Turner, PhD

rturner@plos.org

Please finalize the data statement, adding details for a non-author contact. 

Please adapt the title to better match journal style. We suggest "Variation in HIV care and treatment outcomes by facility in South Africa, 2011-2015: a cohort study". 

Please make the abstract subheadings "Background/Methods and findings/Conclusions". 

Please add a new final sentence to the "Methods and findings" subsection, summarizing 2-3 of the study's main limitations. 

Please begin the "Conclusions" subsection with "In this study, we observed that ..." or similar. 

After the abstract, please add a new and accessible "author summary" section in non-identical prose. You may find it helpful to consult one or two recent research papers published in PLOS Medicine to get a sense of the preferred style. 

Please adapt the final sentence of your Introduction to remove elements of discussion (e.g., "... complements ongoing efforts ..."), which can be moved to the discussion section if you wish. 

At the start of the methods section, should that be "2011-2015"? Similarly, early in the Discussion section (currently "2011-2017"). 

Early in the methods section of your main text, please state whether or not the study had a protocol or prespecified analysis plan, and if so attach the relevant document as a supplementary file (referred to in the text). Please highlight analyses that were not prespecified. 

Please add a sentence on ethics approval to the methods section - e.g., noting that this was not required. 

Please restructure the early part of the discussion section of your main text, so that the first paragraph provides a summary of the study's findings. 

Throughout the text, please move reference call-outs to fall before punctuation (e.g., "... measures of quality [9,21].".

Please spell out the institutional author name for reference 24, and for other references where necessary. 

Please add a completed checklist for the most appropriate reporting guideline, e.g., STROBE or RECORD, referred to in the methods section of your main text. In the checklist, please refer to individual items by section (e.g., "Methods") and paragraph number rather than by line or page numbers, as the latter generally change in the event of publication. 

Comments from the reviewers:

*** Reviewer #1: 

[See attachment]

Michael Dewey

*** Reviewer #2: 

Bor et al. use a large national database with laboratory parameters to design a quality assessment methodology for facilities in South Africa. 

Looking at the huge number of data, complicated by missing values, changing guidelines and varying settings, the authors should be complicated for their effort, especially since the appear to have found an underlying measure of quality of hiv care that correlate with all cascade steps, except CD4 at presentation. 

I do not understand why they picked CD4 at presentation as a quality measure: it doesn't really say anything on how facilities perform with respect to testing; it is more a marker of the population, in particular how they might come forward to medical facilities - it is certainly not an actionable parameter: you cannot invite people to visit to your facility earlier. All other measures are issues where activities of the facilities can really improve the outcomes. The authors should clarify why they chose this in the first place; I am not surprised it didn't turn out as a significant step in their quality measure - I miss in the discussion if they want to keep it in their evaluation.

In the same line: why is CD4 recovery a measure? This is very individually determined, and hardly a factor that a facility can improve (except by viral suppression, which is already a parameter)

The introduction is long and for me it is not entirely clear, why they want to express quality in one score fabricated with a quite complicated statistical approach, where the outcome needs a lot of explanation. The charm of the cascade of care, is that one can see which step is the weakest part of the cascade. Furthermore, if one strives to get one score, why not the product of the several steps: for instance 70%retention x 80%cART x 80%suppression would make 0,45

The authors used the data from over 4000 South African facilities, but showed in their S1 Figure that 2300 facilities were excluded for various reasons. Although I understand their choice, I think they should have discussed in their limitations section what this means for their conclusions, especially since they conclude that larger centers appear to have a higher quality. The data of how many patients were not analyzed this way? How can quality be judged in the facilities that were excluded? Or will the authors now advise that hiv care can better be concentrated in larger facilities?

I miss what the quality score tells an individual facility, since I presume all facilities will get their own score, so they can compare their performance longitudinally, but also with best practices. I expect that the score will be broken down to the individual steps of the cascade, so every facility can better understand which steps need more attention for improvement. 

Finally, I think overall the texts are too long and therefore not always easy to read. I would advise to shorten it, with more focus on the reason for choosing this approach, what individual facilities can do with "one individual score" and how this score is different from the already used cascade of care approach

*** Reviewer #3: 

In this paper, the research team uses laboratory data (CD4+ count, viral load) from the NHLS in South Africa to construct a cohort of HIV+ individuals in eight of nine provinces, then combines a number of quality metrics derived from the lab data into a single quality measure. They then analyse the variation in this quality measure across space and time, and explore population-level and facility-level factors associated with quality. This is really interesting and thought-provoking, and builds on previous work from this group using this constructed lab cohort to provide insight into the national HIV programme. I found some of the main results quite surprising, particularly the apparent increase in quality over time (over a period when services and facilities were becoming very stretched). Whilst there are obvious limitations to analysing laboratory data without linked clinical data, I think these findings do provide some helpful insight and show the potential of the use of data to make inferences about quality of services and to guide decision making. 

Minor comments

1. I would suggest that the exclusion of Western Cape should be mentioned in the abstract so that it's clear this analysis is only based on data from eight provinces

2. I would like to see some explanation for why the data for this analysis stopped in 2015/2016, especially if making an argument that this is something that could be potentially useful in real time to track service quality. Is the work on the lab cohort not ongoing?

3. Methods, p5 (Retention 12-months after first CD4 count <350) - "All retention measures were assigned to the facility where the care episode began." I thought this might be worth more discussion, given the high population mobility in SA - especially where you discuss the rural/urban comparison and where you discuss the possibility of people transferring care in search of better quality care. If this does occur (and it probably does to some extent) then I wonder if this might potentially introduce bias to your analysis. It would be helpful (if possible) to report in the results the proportion of people that had lab results from more than one facility (suggesting move/transfer)

4. Methods, p6 (Viral suppression among patients monitored) - One of the limitations of having only lab data decoupled from clinical data is you can't identify those who are having VL checked when returning to care after LTF. So the LTF could be considered a metric of poor quality but then returning to care (where VL, if checked, will naturally be high) could be considered a metric of good quality care. You have a similar problem with the CD4 recovery metric. Again I recognize you can't do anything about this but just flagging the cycling in and out of care as a substantial limitation that makes inferences about quality from these data quite complicated, and might help to partly explain the low correlation of individual facility-year-level quality measures

5. Methods, p6 (CD4 recovery) - "Some facilities have been slow to implement viral load monitoring and have relied instead on CD4 monitoring as an alternate measure of treatment outcomes." I'm not sure I agree with this statement - yes there have been problems with poor coverage and quality of routine VL monitoring but I don't think that's because people have chosen to use CD4 monitoring instead

6. Table 3 - typo "District or CHC hospital" should be "District hospital or CHC", although should really spell out community health centre either in table or as footnote

7. In the discussion of the limitations of this analysis, I would prefer to see more explicit discussion of why lab data uncoupled from clinical data are imperfect for estimating quality of HIV services. What is it in particular that you miss without linked clinical data and how might this affect these inferences about quality? See comment above about retention and the challenge of people cycling in and out of care for example

Richard Lessells

***

[LINK]

---

## [Decision Letter · Decision Letter 2]

5 Nov 2020

Dear Dr. Bor,

Thank you very much for re-submitting your manuscript "Variation in HIV care and treatment outcomes by facility in South Africa, 2011-2015: a cohort study" (PMEDICINE-D-20-03265R2) for consideration at PLOS Medicine.

I have discussed the paper with editorial colleagues and it was also seen again by two reviewers. I am pleased to tell you that, provided the remaining editorial and production issues are dealt with, we expect to be able to accept the paper for publication in the journal.

[LINK]

Please let me know if you have any questions in the meantime. Otherwise, we look forward to receiving the revised manuscript shortly. 

Sincerely,

Richard Turner, PhD

rturner@plos.org

Requests from Editors:

Please remove one "of" from the second line of the abstract.

Generally, we ask you to add details to your abstract to reflect the methodology and findings of the study more explicitly. Early in the "Methods and findings" subsection of your abstract, we think that "(94% Sen, 99% PPV)" will need some additional explanation, for example. 

Can you add a sentence, say, to the abstract to quote mean facility size, for example, in terms of patients?

We also suggest adding a sentence to quote key elements of the data from table 1, including changes in retention and viral suppression over the study period, say.

Please quote the numerical value, with 95% CI, of the "half a standard deviation" mentioned in your abstract.

Please quote one further study limitation in the final sentence of the "Methods and findings" subsection - e.g., the absence of data from the Western Cape. 

We suggest adding a few words to the final sentence of the abstract to make it less general.

Please mention the absence of data from the Western Cape early in your Methods section, where "all patients" are mentioned.

Please refer to the attached STROBE checklist in your Methods section ("See S1_STROBE_Checklist" or similar). 

Please reword the first sentence of the Discussion section, regarding "complete data" bearing in mind the absence of data from the Western cape, for example. 

Please ensure that all reference call-outs fall before punctuation, e.g., that for reference 28 at the end of the Methods section. 

We generally ask that p values are quoted exactly unless <0.001, and ask that you amend "p<0.01" at the end of the Results section. 

Please review the reference list and remove publisher names (for example from reference 1) and other unnecessary information (e.g., "(80-)" from reference 5, and "[Internet]" from reference 7). 

Please add full access information to reference 3. 

All display items must be publishable under a CC BY licence, and we suspect that this will not be feasible for the current figure 4 - please explain in your revision how you have addressed this issue. 

Comments from Reviewers:

*** Reviewer #1: 

The authors have addressed all my points.

Michael Dewey

*** Reviewer #2: 

The authors have delivered an extensive rebuttal, in which they explained their choices and they furthermore approved the manuscript according to reviewers suggestions, where appropriate

***

[LINK]

---

## [Editor Report · Decision Letter 3]

11 Mar 2021

Dear Dr. Bor,

I am writing concerning your manuscript submitted to PLOS Medicine, entitled “Variation in HIV care and treatment outcomes by facility in South Africa, 2011-2015: a cohort study”.

We have now completed our final technical checks and have approved your submission for publication. You will shortly receive a letter of formal acceptance from the editor.

Kind regards,

PLOS Medicine